# Soil Organic Matter Degradation in Long-Term Maize Cultivation and Insufficient Organic Fertilization

**DOI:** 10.3390/plants9091217

**Published:** 2020-09-17

**Authors:** Jiří Balík, Martin Kulhánek, Jindřich Černý, Ondřej Sedlář, Pavel Suran

**Affiliations:** Department of Agro-Environmental Chemistry and Plant Nutrition, Faculty of Agrobiology, Food and Natural Resources, Czech University of Life Sciences, 165 00 Prague, Czech Republic; kulhanek@af.czu.cz (M.K.); cernyj@af.czu.cz (J.Č.); sedlar@af.czu.cz (O.S.); suranp@af.czu.cz (P.S.)

**Keywords:** C balance, humic acids, fulvic acids, mineral N, farmyard manure, straw

## Abstract

Soil organic matter carbon (C_SOM_) compounds degradation was observed in long-term field experiments with silage maize monoculture. Over a period of 26 years, the content of carbon in topsoil decreased by 22% in control unfertilized plots compared to 25% and 26% in treatments fertilized annually with mineral nitrogen. With annual wheat straw application (together with mineral N), the content of C_SOM_ decreased by 8%. Contrary to that, the annual application of farmyard manure resulted in a C_SOM_ increase of 16%. The ratio of carbon produced by maize related to total topsoil C_SOM_ content ranged between 8.1–11.8%. In plots with mineral N fertilization, this ratio was always higher than in the unfertilized control plots. With the weaker soil extraction agent (CaCl_2_), the ratio of carbon produced by maize was determined to be 17.9–20.7%. With stronger extraction agent (pyrophosphate) it was only 10.2–14.6%. This shows that maize produced mostly unstable carbon compounds. Mineral N application resulted in stronger mineralization of original and stable organic matter compared to the unfertilized control. However, the increase of maize-produced carbon content in fertilized plots did not compensate for the decrease of “old” organic matter. As a result, a tendency to decrease total C_SOM_ content in plots with mineral N applied was observed.

## 1. Introduction

The amount and quality of soil organic matter (C_SOM_) are very significant parameters for assessing soil fertility. C_SOM_ is influenced by a range of soil-climatic conditions, systems of soil management, cultivated crops, fertilization, agrotechnical issues, etc. Agricultural crops differ significantly in the amount and quality of post-harvest residues and root biomass. The crops with high potential to decrease the content of organic matter in soil comprise maize, especially if grown for silage [1]. Loges et al. [2] compared the average amount of post-harvest residues and roots among maize monoculture and rotated crops (clover grass–maize–wheat). The amount of carbon supplied into the soil (post-harvest residues + roots) was 1.3 t ha^−1^ year^−1^ in the case of monoculture, compared to 1.6 t ha^−1^ year^−1^ in crop rotation. As a result of mineralization, the annual carbon decrease was 530 kg C ha^−1^ year^−1^ in monoculture, whereas in crop rotation it was only 120 kg C ha^−1^ year^−1^.

C_SOM_ can be divided into stable and labile fractions. Stable carbon forms are represented by humic acids (C_HA_), fulvic acids (C_FA_), and humins (C_HU_). Carbon sequestration in terms of C_HA_, C_FA_, and C_HU_ content and quality is very important to understand soil quality degradation [3,4].

Gregorich et al. [5] found that after seven years of maize cultivation as a monoculture, the ratio of carbon originating from maize (C_4_) related to C_SOM_ (C_4_/C_SOM_) was only 15% of total C_SOM_. The previous crop was permanent grass stand. Collins et al. [6] determined the C_4_ ratio in C_SOM_ as 23–60% in fertilized plots and 9–32% in unfertilized control plots. These results were obtained in the main maize-growing regions in the USA in fields with maize cultivated for 8–35 years. Similarly, [7,8] reported higher C_4_ ratios in fertilized plots compared to unfertilized ones; this was affected mainly by higher biomass yields (including root biomass) in the fertilized crop. Bettina et al. [9] evaluated experiments lasting 40 years with maize monoculture, which followed after rye monoculture. The C_4_ ratio in C_SOM_ was 9.5% and 14.1% in unfertilized crops and fertilized crops in topsoil, respectively; the values for subsoil were 5.7% (no fertilization) and 7.2% (fertilization). Bettina et al. [9] presumed that the main reason for the lower C_4_ ratio compared to other studies was the growing of maize for silage in their experiments. Almost all the biomass was thus harvested and removed from the fields and only low stubble remained. Similarly, low C_4_ ratios were also reported by [10,11].

Maize biomass carbon is easily mineralized in the soil; hence, the increased C_4_ ratio is determined mostly in easily hydrolysable fractions (C_DOC_). Depending on the purpose of cultivation (silage/grain), length of monoculture cultivation, a system of fertilization, soil-climatic conditions, the C_4_ ratio in C_DOC_ was almost 1/3 [5,9,12]. Fertilized plots also had higher total content of C_DOC_ [13]. Even higher values than in C_DOC_ were observed for C_4_ in microbial biomass carbon [5,9]), again with the values of fertilized plots being higher. The results also show that “younger” organic matter (plant residues) is mineralized more quickly than “older” organic matter in the soil. C_4_ thus accounts for the main share of carbon loss in the form of CO_2_; Bettina et al. [9] reported almost 80.0%.

The trials of Loges et al. [2] suggest the ratio of root biomass and exudates as 18% compared to the aboveground biomass. Out of the C amount in roots and post-harvest residues, the main part remains in the topsoil and a lesser amount in subsoil [9,14,15,16]. Loges et al. [2] presumed that it was sufficient to examine the 0–30 cm soil layer to determine the impacts of different cultivation systems on post-harvest residues’ C input into soil. Rasse et al. [10] found almost all C_4_ in the depth 0–90 cm, 63% of it being in the 0–30 cm layer. In the subsoil, the content of C_SOM_ is significantly lower and more stable [17]. The reason is a tighter relationship of organic matter to clay particles [18] and also a lower air content, i.e., lower mineralization intensity.

The aim of this paper was to assess the changes in C_SOM_ amount and quality in long-term silage maize monoculture, as well as compare the effects of different mineral and organic fertilizers. This research is also important from the viewpoint of long-term sustainable soil management in the Czech Republic (CR), with respect to increasing usage of maize in biogas plants [19], low intensity of organic fertilization of maize and other crops, and unbalanced C in agricultural systems. These issues are very important, as shown by the high ratio of maize in crop rotations. Moreover, maize is often grown for several consecutive years without rotation. In the CR, its acreage is 10.5% of agricultural soils—about 232,400 ha being silage maize + 74,800 ha for grain [20]. Furthermore, luvisol (the soil type chosen for our experiment) is representing 4.29% of the area of agriculturally used soils in the CR, which means 179,167 ha [21].

## 2. Results

The explanation of abbreviations is mentioned above in the list of abbreviations and further in the chapter materials and methods.

### 2.1. Yield Parameters

Differences in yields of harvested biomass over a period of 26 years are shown in Figure 1. Although the differences among fertilized treatments are relatively high (sometime even 18%), statistically significant differences were not observed. This is mainly due to the variability of climatic conditions during experimental seasons, where the effect of individual fertilization varied according to related season.

The values in fertilized plots reached 12.2–14.0 t DM ha^−1^ and correspond to site conditions. The highest yields were obtained in the urea ammonium nitrate (UAN) + straw (St) treatment (Figure 1) with the highest N dose applied, i.e., 153.5 kg N ha^−1^ year^−1^ (120 + 33.5) (see the chapter methodology).

### 2.2. Carbon Produced by Maize

Table 1 summarizes the maize carbon (C_4_) results. The obtained average yields of dry matter and its C content were used to calculate the amount of carbon transported every year from the field. The yield of 8.84 t DM ha^−1^ also represents 3755 kg C ha^−1^. The amount of carbon in post-harvest residues (stubble) was determined experimentally in 2018. The total carbon accumulated in the aboveground biomass was calculated as the sum of “harvest” + “stubble” values. Similarly, the carbon amount in topsoil roots was obtained by analyses in 2018 (at the harvest period). To compare our experimentally determined results, the values calculated based on various algorithms reported in the literature are presented in Table 1.

### 2.3. The Changes of Soil Organic Matter Carbon (C_SOM_) Contents

In 1993, the amount of C_SOM_ in the topsoil was 1.26%. After harvest in 2018, the following values were determined: control 0.98%, ammonium sulfate (AS) 0.93%, UAN 0.95%, UAN + St 1.16%, manure 1.49% (Figure 2). There was a significant decrease of C_SOM_ in the topsoil of all treatments, except for farmyard manure. In relative values, the C_SOM_ content decreased by 22% in the control and mostly in AS by 26%. C_SOM_ content in subsoil was not determined at the beginning of the experiment, but Figure 2 clearly shows a lower C_SOM_ content in the subsoil in the AS treatment compared to other treatments. On the contrary, treatment with farmyard manure resulted in significantly higher C_SOM_ content also in the subsoil, which is probably related to rather deep tillage (28 cm).

### 2.4. Carbon Balance

Table 2 presents the score of carbon losses as a result of mineralization as well as respiration of CO_2_. Column 2 shows the calculated difference of C_SOM_ content at the beginning of the experiment (1993) and after harvest in 2018. For example, the loss in the AS treatment was−571 kg C ha^−1^ year^−1^. Further, the C amount supplied with organic fertilization was calculated, e.g., for UAN + St treatment it was 2140 kg C ha^−1^ year^−1^.

The data concerning the carbon amount ploughed into the soil in the form of post-harvest residues (stubble) and roots (topsoil only) originate from our analyses in 2018. It was evident that the highest carbon losses are in the treatment UAN + St, i.e., 3415 kg C ha^−1^ year^−1^. These results confirm the dynamics of straw mineralization in soil. The losses in the treatment with farmyard manure were 2333 kg C ha^−1^ year^−1^. Manure contains organic matter, which is significantly more stable compared to straw and affects C_SOM_ stability considerably. Presuming that stubble residues do not significantly affect C content in soil and are mineralized quickly (about three times faster than manure) and further that root biomass needs a similar time to decompose as manure, about 16% transformation of carbon from manure to stable soil compounds may be deduced. The residual amount then mineralizes or may counterbalance the losses by mineralization of the original organic matter.

Methods using stable carbon isotopes (^13^C/^12^C) allowed us to determine the ratio of maize carbon (C_4_) in the total topsoil and subsoil C_SOM_. The reported analyses were assessed only for the unfertilized control plots and for mineral N treatments. In the fertilized plots, the content of maize carbon in total C_SOM_ in topsoil and subsoil increased. In the topsoil of the control plots, the C_4_ ratio in C_SOM_ was 8.1%, in the case of AS it was 10.3%, and in UAN 11.8%, respectively (calculated as C_4_/(C_3_ + C_4_) from Table 3. * 100%). The C_4_ ratio in C_SOM_ in subsoil was about 25–50% lower compared to topsoil (control 5.5%, AS 7.9%, UAN 6.0%).

Table 3 shows statistically conclusive differences in the content of “old” organic matter (C_3_) among control and N fertilization treatments. In the case of AS and UAN treatments, stronger mineralization of original and stable C_SOM_ occurred compared to the control. Contrariwise, N mineral fertilization resulted in a significant increase in C_4_ content. This process also occurs in the subsoil, yet it is less dynamic. The values reported in Table 3 further demonstrate that mineral N fertilization alone increases hydrolysis of C_SOM_, which is manifested by a significant increase of extractable organic compounds. C_CaCl2_ represents the carbon determined with the weak extracting agent and C_PF_ with strong one, respectively. It was evident that C_4_ comprises mostly unstable and easily hydrolysable forms. The ratio of C_4_ in C_CaCl2_ was 17.9–20.7%; in C_PF_ it was 10.2–14.6%.

### 2.5. The Soil Organic Matter Quality Parameters

The relationship between the amount of extractable carbon and total C_SOM_ is shown in Figure 3. The highest values were obtained in the AS treatment. The lowest ratios of C_CaCl2_ and C_PF_ to C_SOM_ were always obtained in the control and farmyard manure treatments, which suggests higher stability of organic matter in soil compared to other treatments. High values in the UAN + St treatment confirm previous results of rather fast mineralization of straw in soil (Table 2). Figure 4 shows the total amount of extractable carbon using CaCl_2_ or Na_4_P_2_O_7_. Considering the high total C_SOM_ content in the farmyard manure treatment, high carbon values were also obtained at the extraction with a strong solution of Na_4_P_2_O_7_.

Over the period of 26 years, a significant decrease of the C_HA_ content in the control and AS plots (by 50%) was observed. By contrast, the farmyard manure treatment showed significantly higher C_HA_ compared to other treatments. A similar tendency may be observed in case of the C_HA_/C_FA_ ratio (Figure 5).

### 2.6. The Quality of Post-Harvest Residues

Mineral N fertilization destabilized organic matter in the soil. This was demonstrated by values of the C_SOM_/N_t_ ratio in topsoil and subsoil (Figure 6). In topsoil, a significant decrease in the C_SOM_/N_t_ ratio was observed in AS and UAN treatments compared to the control. The reason for the change in this ratio is mainly a decrease of C_SOM_ content in mineral N fertilization treatments, because N_t_ content was the same in all three compared treatments (0.10%). In the subsoil, this difference was not significant, but there was a real tendency to its decrease. It may be presumed that during subsequent years the differences will also become more significant. The highest C_SOM_/N_t_ ratio was always observed in the farmyard manure treatment.

The reason for the C_SOM_/N_t_ ratio change may be inferred from the increased mineral N content in soil and, furthermore, from the quality of the tilled post-harvest residues (stubble) and root biomass (Figure 7). In stubble, the carbon/nitrogen (C/N) ratio in control plots was 315/1, in AS it was 107/1, and in UAN 88/1, respectively. The lowest (70/1) value was determined in the UAN + St treatment, which corresponds to the highest N fertilization intensity within this treatment. Similarly high differences among the treatments were observed in the analysis of roots in topsoil; the highest ratio was also observed in the control treatment (C/N = 123/1), whereas it was very low in the mineral N treatments (C/N = 49/1). These differences reflect the fact that both root biomass and plant residues were more stable in control plots compared to N fertilization.

Figure 8 demonstrates that this was due to post-harvest residues and roots that supplied 90–100% more carbon in mineral N treatments compared to the control. The highest amount of post-harvest residues was in the farmyard manure treatment (stubble: 360 kg C ha^−1^ + roots: 768 kg C ha^−1^ = 1128 kg C ha^−1^).

## 3. Discussion

The luvisols comprise over 500–600 million ha worldwide and are situated mainly in temperate regions such as in the East European Plains and parts of West Siberian Plain, the North East of the United States of America and Central Europe, but also in the Mediterranean region and southern Australia. Most luvisols are fertile soils and suitable for a wide range of agricultural uses [22]. The worldwide area used for maize forage is about 16.8 million ha [23]. Our results, which are presenting the soil carbon transformations in the maize monoculture cropped on luvisol also provide information that could be useful not only for regional purposes. Similar results can be expected in similar soil conditions by cropping the silage maize in long-term monoculture.

The average yield of the harvested biomass in the control plots over a period of 26 years was 8.84 t DM ha^−1^ year^−1^, which represents 3755 kg C ha^−1^ year^−1^. In fertilized plots, the biomass yield was 40–58% greater compared to the control. Although the intensity of mineral nitrogen fertilization (120 kg N ha^−1^ year^−1^) corresponded to the yields obtained and from the long-term perspective, it was in compliance with N uptake by plants, mineral N resulted in a C_SOM_ decrease in soil. During the experimental period (26 years), the most significant decrease was that of C_SOM_ in the AS treatment (26%) and the UAN treatment (25%). It is almost alarming that in an annual application of wheat straw (5 t DM ha^−1^ = 2140 kg C ha^−1^) C_SOM_ also decreased (by 8%). It is obvious that straw is relatively quickly mineralized after its incorporation into the soil [24,25,26,27]. Due to this fact it is clear that cereal straw application itself cannot improve the soil organic matter content and quality. This was confirmed in our experiments by the aforementioned decrease of C_SOM_ at this treatment.

The annual average dose of manure was 18.7 t ha^−1^, which was above average, but not extremely high. Cattle manure with a C/N ratio of 13.4/1 was used. The relatively low C/N ratio in manure fully corresponds with stable technologies currently used in CR. This means a decrease in the C/N ratio compared to the past when the standard values were 15–18/1. However, manure application contributed to a significant increase in the C_SOM_ values (from 1.26% to 1.46%). A significantly positive effect of manure may also be observed in the study of Menšík et al. [4]. Schmidt et al. [28] reported an increase of C_SOM_ values by 32% (from 1.24% to 1.64%) at a dose of 12 t ha^−1^ year^−1^ and a period of 50 years for rye cultivated in a monoculture. The importance of the C/N ratio is evident, but the quality of the incorporated organic matter is important as well. For example, in the UAN + St treatment, the resulting ratio in the fertilizers applied, C/N = 14.6/1, was higher compared to manure, but it did not result in a C_SOM_ increase. Manure contains significantly more stable organic matter than straw and thus helps to control C_SOM_ stability.

Furthermore, in the control plots, a decrease of C_SOM_ content was observed, namely by 22%, which corresponds to losses of 485 kg C ha^−1^ year^−1^. In the case of the AS treatment, this represents 571 kg C ha^−1^ year^−1^ compared to UAN with 537 kg N ha^−1^ year^−1^. These values do not include C losses due to mineralization of post-harvest residues. The values are high but fully correspond with the results of [2] concerning long-term silage maize cultivation. These authors reported average losses of 530 kg C ha^−1^ year^−1^ in an unfertilized plot and 550 kg C ha^−1^ year^−1^ in a fertilized one. A decrease of C_SOM_ content in maize monoculture was also documented by [7,9]. A C_SOM_ content increase was reported mostly in maize cultivated for grain [6,10]. For example, Liang and MacKenzie [29] recorded increased C_SOM_ content in topsoil by 18% after six-year maize cultivation for grain. Basically, silage maize cultivation results in a small amount of aboveground post-harvest residues (stubble). In the present study, the height of stubble was about 10 cm. The carbon content in stubble ranged between 208 kg C ha^−1^ (control) up to 360 kg C ha^−1^ (manure). This is in good agreement with the assessment of [15] who reported a value of 290 kg C ha^−1^.

To evaluate C balance, it is necessary to determine the number of roots. In our experiments, the amount of carbon in roots was determined after harvest. Determined values were in a range of 394 kg C ha^−1^ year^−1^ (control) up to 809 kg C ha^−1^ year^−1^ (UAN + St). Balesdent and Balabane [15] determined the number of roots at the level of 12% of the aboveground biomass. Using this model for calculations, determined values were in the range of 476 kg C ha^−1^ year^−1^ (control) up to 738 kg C ha^−1^ year^−1^ (UAN + St). The aforementioned calculations considered the whole root biomass, including subsoil. The experiments of [2] demonstrated a ratio of root biomass and root exudates of 18% compared to aboveground biomass. Using this model of calculation, the determined values were in the interval from 870 kg C ha^−1^ year^−1^ (control) up to 1349 kg C ha^−1^ year^−1^ (UAN + St). The values for the topsoil and subsoil calculated according to Rasse et al. [10] were as follows: 63% topsoil, 37% subsoil. The comparison of calculated values and those obtained in our measurements show good agreement (Table 1). Both methods show the lowest values in the control and the highest in the UAN + St treatment. To conclude, the values determined by measurements were about 10% lower than the calculated ones. There are several possible reasons: (i) the amount of roots was determined after 26 years of monoculture. As reported by [28], in the case of long-term monoculture, the amount of roots and rhizodeposition decrease by as much as 30%; (ii) the calculations by [2] also comprise exudate carbon that was not determined in our experiments; and (iii) the highest difference among the calculated and measured values was observed mostly in the control; in this variant, a significant tendency to lower yields of aboveground biomass and, thus, roots were observed over the years. For calculation models, the average biomass yield value over the period of 26 years was used.

Studying the root biomass in field conditions always involves error, but it is possible to assess 30 the C ratio using the values determined directly in our study (Table 2). It is evident that the amount of post-harvest residues in silage maize is very small (stubble: 208–360 kg C ha^−1^; roots 394–809 kg C ha^−1^) and cannot replace the losses due to C_SOM_ mineralization [2]. In post-harvest residues, the C/N ratio was observed (Figure 7). The results obtained prove that root biomass as well as the stubble residues were more stable (higher C/N ratio) in control plots than in mineral N-fertilized ones. Additionally, the manure treatment showed a statistically significant increase in the C/N ratio compared to the mineral N fertilization. Similar to our experiments, [2] also identified a significantly higher C/N ratio in stubble (197/1) compared to roots (28/1).

N applied in mineral fertilizers showed a significant decrease in the C_SOM_/N_t_ ratio in topsoil, which confirms the hypothesis that mineral N fertilization leads to soil organic matter (SOM) destabilization. Contrariwise, manure treatment showed an increase in this ratio.

The overall C ratio (Table 2) clearly shows that the lowest carbon losses due to mineralization were recorded in the control (1087 kg ha^−1^ year^−1^). Contrariwise, the highest losses were determined in the UAN + St treatment—3415 kg C ha^−1^ year^−1^. The evaluation of carbon losses due to the mineralization process related to the total carbon amount in the harvested biomass is very interesting. The results clearly show that the control (28.9%), AS (29.7%), and UAN (28.4%) treatments do not differ significantly. The application of organic fertilizers turns this ratio significantly more negative: UAN + St (57.6%) and manure (41.1%) (Table 2). High losses of C in soil highlight the importance of this research and at the same time the necessity of a complex approach to its solution.

The relationship between the amount of extractable carbon (C_CaCl2_, C_PF_) and the total amount of C_SOM_ documents the fact that the control plots and farmyard manure treatment had more stable organic matter and consequently, a lower intensity of mineralization [29]. The highest values were recorded in the AS treatment—the ratio of C_CaCl2_/C_SOM_ was 0.11% and that of C_PF_/C_SOM_ 28%. Similarly, a very low extraction strength (0.01 mol L^−1^ CaCl_2_) was also determined in our previous experiments at different sites [30]. In contrast, a lower ratio of C_PF_/C_SOM_ was recorded in our trials compared to the results of Ellerbrock and Kaiser [31]. In their trials, about 40% of C_SOM_ was extracted while in our trials it was only 22–28%. Furthermore, [4] used the C_PF_/C_SOM_ ratio to assess the stability of SOM, suggesting that the higher this ratio is, the less stable the C_SOM_. Similar to our experiments, [4] also determined the lowest value in the manure treatment (27.6%) and the highest in the mineral nitrogen, phosphorus and potassium (NPK) treatment (34.7%). Over the period of 26 years, C_SOM_ degradation occurred in almost all treatments, except farmyard manure. Both the content of C_HA_ and the C_HA_/C_FA_ decreased. This process was the fastest in the AS treatment. It is evident that manure helps to increase the content of C_HA_ as well as its ratio in C_HS_ [4]. In the UAN + St treatment, the content of C_HA_ did not increase; the C_HA_/C_FA_ did not improve, even with the application of high doses of wheat straw.

Using the analyses of ^13^C/^12^C allowed us to determine the ratio of C_4_ in the C_SOM_ and C_4_ ratio in different extractable fractions. This was realized only for the mineral fertilizing treatments for two reasons: (i) the ^13^C/^12^C was not evaluated at organic inputs (FYM, straw), because we have not the data for ^13^C/^12^C in input materials; (ii) the results originate from the long-term experiments with maize monoculture, which allows us to make a reliable evaluation. It was evident that the C_4_ ratio was higher in the fertilized plots compared to the control (control 8.10%, AS 10.3%, UAN 11.8%). The higher ratio in fertilized plots is reported by many authors [6,7,8,9]. However, the C_4_ ratios determined in these studies differ significantly. There is a significant influence of the monoculture duration—from six years [11], up to 40 years [9]. The purpose of cultivation (grain/silage) is also very important. Our values are in good compliance with [9], who reported a C_4_ ratio in C_SOM_ of 9.5% in the unfertilized control and 14.1% in the fertilized treatments. Our experiments and the work by [9] focused on silage maize, where almost all aboveground biomass is taken off the field. A similarly low C_4_/C_SOM_ ratio (13%) was also determined by [10]. Ellerbrock and Kaiser [31] also determined 14.2%. By contrast, [6] published results where C_4_ represented 60% of C_SOM_ in topsoil. However, it was in long-term grain-maize cultivation. Our results show that the C_4_ ratio in the C_SOM_ of the subsoil is significantly lower. In the control, it was 5.5%, in AS 7.9%, and in UAN 6.0%, respectively. Similarly low C_4_ ratios in C_SOM_ (5–10%) were recorded for subsoil in the studies of [8,9].

A significantly higher C_4_ ratio compared to C_SOM_ was recorded in the hydrolysable fractions of organic compounds (C_DOC_). The 0.01 mol L^−1^ CaCl_2_ extraction solution had a C_4_ ratio in C_CaCl2_ in the interval of 17.9–20.7% and the 0.1 mol L^−1^ Na_4_P_2_O_4_ extractable fraction was in the interval of 10.20–14.60% (C_4_ ratio in C_PF_). It was evident that the weaker the extraction agent, the higher the ratio of maize carbon. Similarly, Ellerbrock and Kaiser [31] determined a ratio of 22% C_4_ for water extractable carbon and 16% C_4_ for C_PF_. Higher ratios than those determined in our experiment have been reported, e.g., by [5] ratio 34% C_4_ in C_DOC_; [12] ratio 30%; [9] ratio 5–30%. A very high ratio of C_4_ in C_DOC_ (29–54%) was reported by [5]; however, it was in long-term grain-maize cultivation. In our experiments, the ratio of C_4_ in C_DOC_ was always higher in fertilized plots compared to the control with no fertilization, which complies with previous results. The results further show that fertilization increases the amount of easily extractable and thus less stable C fraction [13].

Statistically conclusive differences in the content of C_3_ between control and N-fertilized plots document the fact that AS or UAN applications result in stronger mineralization of original and stable organic matter compared to a non-fertilized control (Table 3). The increase of C_4_ content in these treatments does not compensate for the decrease of C_3_ content; the result is a tendency to decrease total C_SOM_ content [2,7]. This process also takes place in the subsoil, but less dynamically. Fertilized plots in our experiments had about 9% lower C_3_ content in topsoil.

## 4. Materials and Methods

### 4.1. Field Experiments

Long-term stationary field trials with maize monoculture were set up at the Czech University of Life Sciences experimental site Červený Újezd in 1993. Maize was grown for silage. The characteristics of the site are shown in Table 4.

The size of individual experimental plots was 170 m^2^ (20 m length × 8.5 m width) and the size of harvested plots 20 m^2^ (two middle rows of the experimental plot). Each treatment (experimental plot) was conducted in four replications in a randomized design. Nitrogen was applied in the same dose (120 kg N ha^−1^ year^−1^) using ammonium sulfate (AS), urea ammonium nitrate (UAN) and farmyard manure (FYM), except for the unfertilized control plots (Cont) and treatment UAN with straw (UAN + St) (Table 5). Nitrogen fertilizers were applied in spring, before maize sowing. In the UAN + St treatment, wheat straw (5 t dry matter (DM) ha^−1^) was applied just before the autumn tillage. Similarly, FYM was applied in the autumn and immediately incorporated with ploughing to minimize nitrogen losses. The amount of FYM corresponded to 120 kg N ha^−1^ (always according to FYM nitrogen content analysis).

All aboveground biomass was harvested and removed from the harvested plot, except for about 10 cm stubble. The roots were extracted by the sampling of topsoil blocks (area 40 × 40 cm; 30 cm depth; 4 subsamples per plot) and were subsequently washed and separated.

Soil samples (topsoil from 0–30 cm and subsoil 30–60 cm depth) were taken up after maize harvest from twelve sampling points per plot, subsequently, mixed, sieved through 5 mm mesh and frozen. To the further described soil analysis, samples from the years 1993 and 2018 were thawed, air-dried, sieved through 2 mm mesh and analyzed (except for CaCl_2_ extraction, where the fresh soil samples were analyzed). The C_SOM_ content was determined in the samples taken up in both years (1993 and 2018). The remaining soil analysis proceeded only in the samples taken up in 2018.

### 4.2. Analysis

A solution of 0.01 mol L^−1^ CaCl_2_ (C_CaCl2_) was used (1:10, w/v) to determine one of the less stable soil C_CaCl2_ fractions. The content of C_CaCl2_ was determined in fresh soil samples by segmental flow-analysis using the infrared detection on a Skalar^plus^System (Skalar, Breda, Netherlands).

Fractionation of humic substances (C_HS_) was undertaken according to the [34] method to obtain the pyrophosphate extractable fraction (C_PF_), which represents the sum of the carbon in humic acids (C_HA_) and fulvic acids (C_FA_). In brief, C_HA_ and C_FA_ were extracted from a 5 g soil sample with a mixture of 0.1 mol L^−1^ NaOH and 0.1 mol L^−1^ Na_4_P_2_O_7_ (1:20, *v*/*v*) solution. Carbon of humic substances C_HS_ and C_HA_ was determined by an oxidimetric titration method. The content of C_FA_ was calculated as the difference between C_HS_ and C_HA_. The degree of humification was calculated as the ratio of C_HS_ content and C_SOM_ content multiplied by 100 [4].

Stable isotope δ^13^C analyses were performed by flash combustion in a Fisons 1108 elemental analyzer coupled with an isotope ratio mass spectrometer Delta V Advantage (ThermoFisher, Bremen, Germany) in a continuous flow regime. The sample size was adjusted to contain a sufficient amount of carbon. Results are reported as δ^13^C values (in per mil ‰) relative to Vienna Pee-Dee Belemite (V-PDB). International standards NBS 22 (−30.031 ‰), IAEA-CH-7 (−32.151 ‰), and in-house standard soil (−27.80 ‰) were used as reference material. The long-term reproducibility was better than 0.15‰. ^13^C/^12^C isotope ratios were expressed as δ ^13^C values.
δ^13^C (‰) = [(δ_sam_/δ_std_) − 1] × 10^3^,(1)
where δ_sam_ = ^13^C/^12^C ratio of sample, and δ_std_ = ^13^C/^12^C ratio of the reference standard (PDB). The δ^13^C values of dissolved organic carbon (DOC) were determined by the method of Buzek et al. [35]. In brief, the solution was acidified with diluted phosphoric acid to remove bicarbonates, and further concentrated by evaporation (at 50 °C) for DOC δ^13^C measurements (Fisons 1108 and Delta V with NBS 22 as internal reference).

The content of total organic carbon and nitrogen in air-dried soils, in farmyard manure, and in plants was determined using oxidation on a CNS analyzer (Elementar Vario Macro, Elementar Analysensysteme, Hanau-Frankfurt am Main, Germany).

Data were processed using basic tests for normal distribution and subsequently one-way analysis of variance (ANOVA, Tukey test; *p* < 0.05) using the STATISTICA 12 (Dell–StatSoft Inc., Austin, USA) program.

## 5. Conclusions

Soil organic matter carbon (C_SOM_) compounds’ degradation was observed in long-term field experiments with silage maize monoculture. Over a relatively short period of 26 years, the content of carbon in topsoil decreased by 22% in control plots compared to 26% in treatments fertilized with mineral N. It is almost alarming that in an annual application of wheat straw (5 t DM.ha^−1^) C_SOM_ also decreased (by 8%). It is obvious that straw was relatively quickly mineralized after its incorporation into the soil. By contrast, the application of farmyard manure resulted in a C_SOM_ increase of 16%.

The ratio of carbon produced by maize from total topsoil C_SOM_ content ranged between 8.1–11.8%. In plots with mineral N fertilization, this ratio was always higher than in the unfertilized control plots. The weaker the soil extraction agent, the higher ratio of maize carbon, which shows that maize produced mostly unstable carbon compounds. Furthermore, mineral N application resulted in stronger mineralization of original and stable organic matter compared to the unfertilized control. However, the increase of maize carbon content in fertilized plots did not compensate the decrease of “old” organic matter. As a result, a tendency of decreasing total C_SOM_ content in plots with mineral N applied was observed.

Generally, it is obvious that silage maize monoculture cropping in a relatively short time period is significantly decreasing the C_SOM_ content as well as its quality expressed by decreasing: (i) content of humic acids and (ii) ratio of humic/fulvic acids.

## Figures and Tables

**Figure 1 plants-09-01217-f001:**
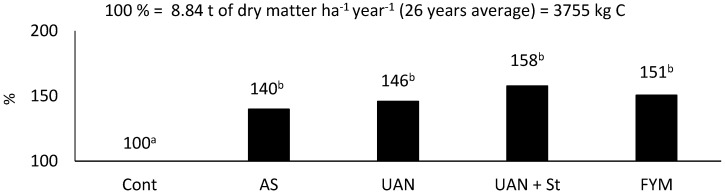
Average maize biomass yield; different letters behind the values are meaning significant differences among investigated treatments (Tukey test; *p* < 0.05); number of replications per treatment *n* = 4.

**Figure 2 plants-09-01217-f002:**
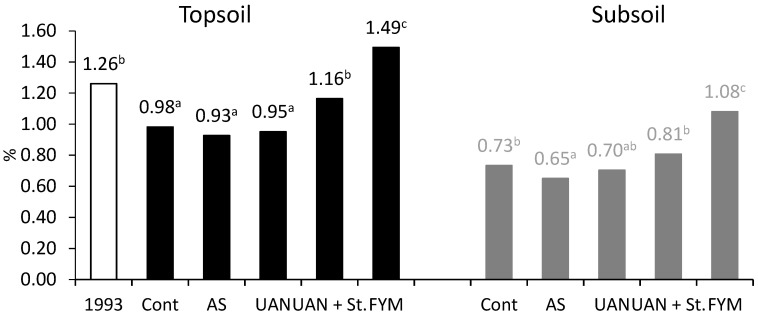
Soil organic matter carbon (C_SOM_) content in topsoil and subsoil; different letters behind the values are meaning significant differences among investigated treatments (Tukey test; *p* < 0.05); number of replications per treatment *n* = 4.

**Figure 3 plants-09-01217-f003:**
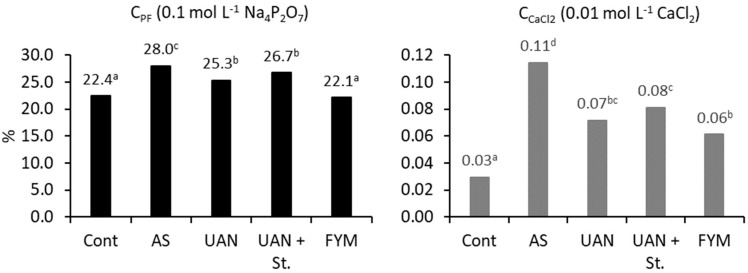
Ratio of extractable carbon of total soil organic matter carbon (C_SOM_); topsoil; different letters behind the values mean significant differences among investigated treatments (Tukey test; *p* < 0.05); number of replications per treatment *n* = 4.

**Figure 4 plants-09-01217-f004:**
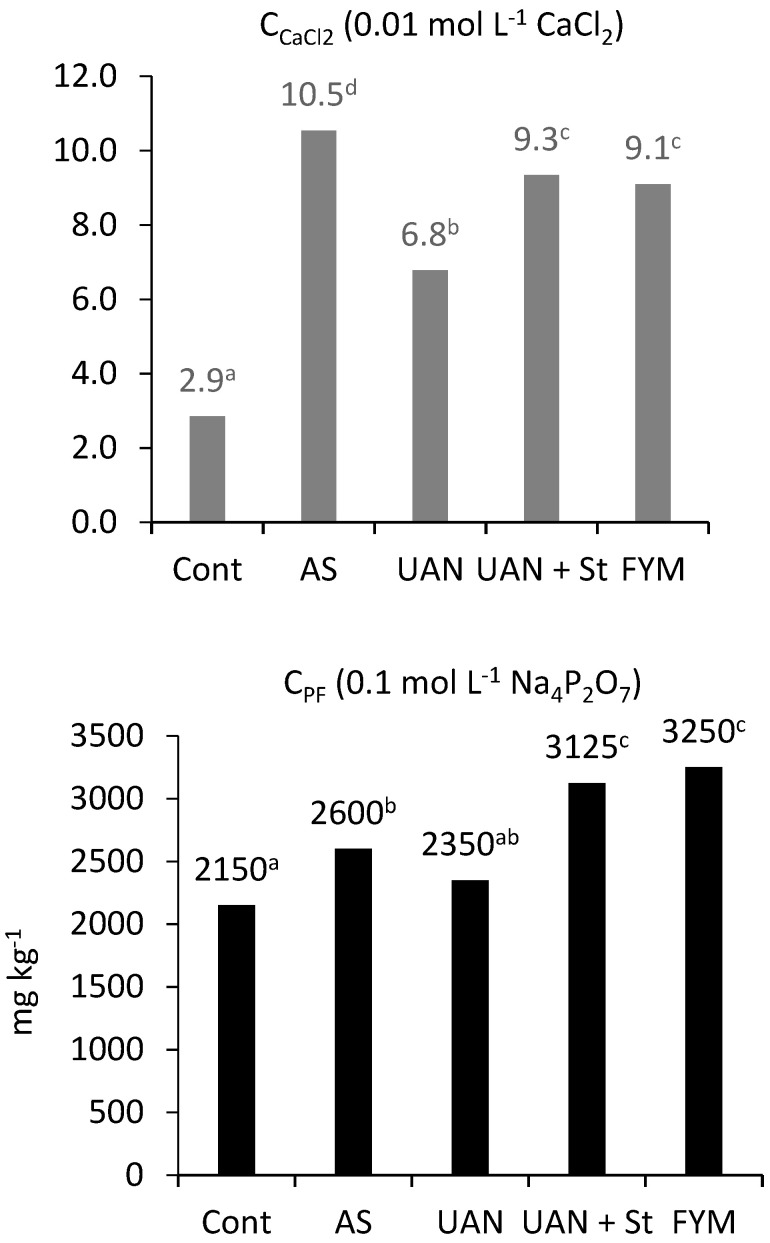
Content of extractable carbon determined with different extraction procedures; topsoil; different letters behind the values mean significant differences among investigated treatments (Tukey test; *p* < 0.05); number of replications per treatment *n* = 4.

**Figure 5 plants-09-01217-f005:**
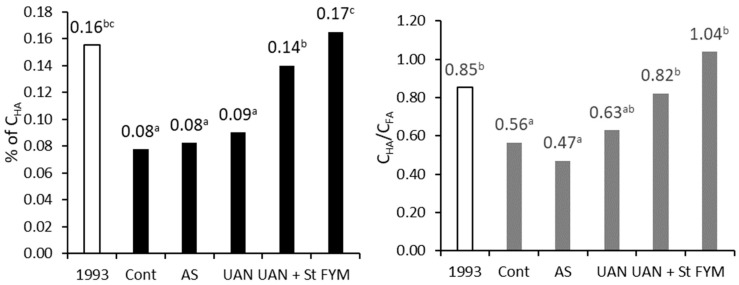
Content of humic acids (C_HA_) and their ratio to fulvic acids (C_FA_); topsoil; different letters behind the values mean significant differences among investigated treatments (Tukey test; *p* < 0.05); number of replications per treatment *n* = 4.

**Figure 6 plants-09-01217-f006:**
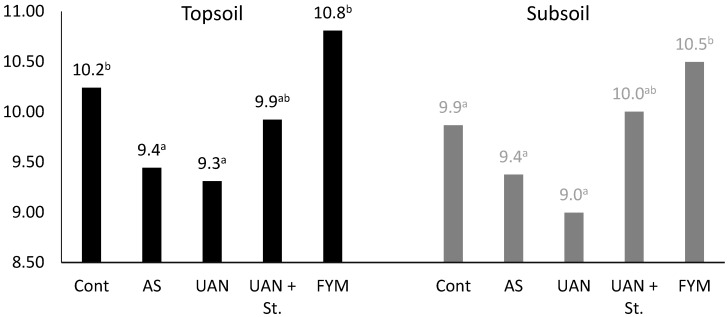
Soil organic matter carbon (C_SOM_)/soil total nitrogen (N_t_) ratio in topsoil and subsoil; different letters behind the values mean significant differences among investigated treatments (Tukey test; *p* < 0.05); number of replications per treatment *n* = 4.

**Figure 7 plants-09-01217-f007:**
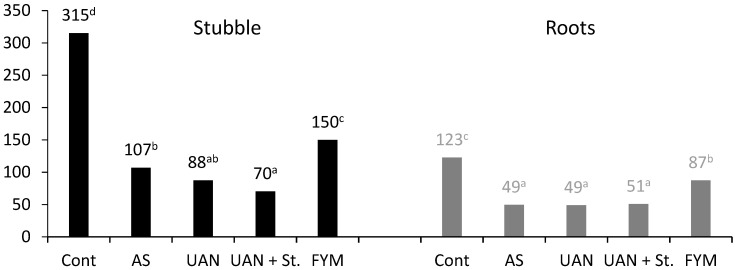
Carbon/nitrogen (C/N) ratio in crop residues of maize; different letters behind the values mean significant differences among investigated treatments (Tukey test; *p* < 0.05); number of replications per treatment *n* = 4.

**Figure 8 plants-09-01217-f008:**
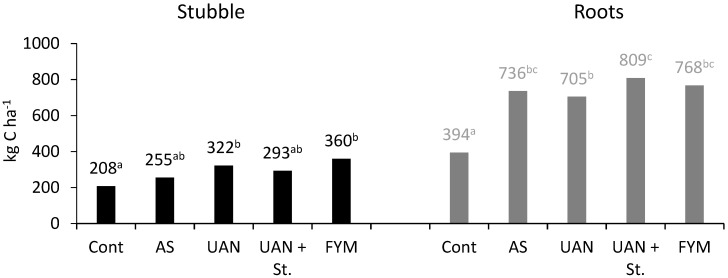
Carbon contents in crop residues of maize; different letters behind the values mean significant differences among investigated treatments (Tukey test; *p* < 0.05); number of replications per treatment *n* = 4.

**Table 1 plants-09-01217-t001:** Carbon balance of maize (kg C ha year^−1^); AS–ammonium sulfate, UAN–urea ammonium nitrate, UAN + st–urea ammonium nitrate + straw, FYM–farmyard manure.

Treatment	Above Ground Biomass				Roots + Exudates ^3^
	Transport by Harvest ^1^ Stubble ^2^		∑	Roots ^2^	Loges et al. [2] Rasse et al. [10]
Control	3755	208	3963	394	548
AS	5210	255	5466	736	756
UAN	5421	322	5743	696	794
UAN + St	5854	293	6147	809	850
FYM	5615	360	5975	768	826

^1^ Average value of 26 year period; ^2^ Determined in the year 2018; ^3^ Calculated according to Rasse et al. [10] and Loges et al. [2].

**Table 2 plants-09-01217-t002:** C_SOM_ loses due to the mineralization (soil organic matter + fertilizer + crop residues) (in kg C ha^−1^year^−1^); AS–ammonium sulfate, UAN–urea ammonium nitrate, UAN + st–urea ammonium nitrate + straw, FYM–farmyard manure, C–Carbon.

Treatment	C in Topsoil ^1^	C from Org. Fertilizer	C in Topsoil + C from Org. Fert.	C in Stubble ^2^	C in Roots ^2^	Total C	Loses/Harvest (%) ^3^
Control	−485	0	485	208	394	1087	28.9
AS	−571	0	571	255	736	1562	29.7
UAN	−537	0	537	322	696	1555	28.4
UAN + St	−173	2140	2313	293	809	3415	57.6
FYM	398	1603	1205	360	768	2333	41.1

^1^ Calculated as difference among C_SOM_ at the beginning of experiment (1993) and year 2018; ^2^ determined in 2018; ^3^ Ratio of C loses compared with carbon content in biomass of harvested maize.

**Table 3 plants-09-01217-t003:** Carbon content in topsoil and subsoil; AS—ammonium sulfate, UAN—urea ammonium nitrate, C—carbon, C_SOM_—soil organic matter carbon, C_CaCl2_—carbon determined with 0.01 mol L^−1^ CaCl_2_, C_PF_—carbon determined with 0.1. mol L^−1^ Na_4_P_2_O_7_, C_3_—“old” soil organic matter, C_4_—carbon from maize.

Treatment	Soil Depth (cm)	Origin of C	C_SOM_ (g m^−2^)	C_CaCl2_ (g m^−2^)	C_PF_ (g m^−2^)
Control	0–30	C_3_ + C_4_	4322 ^a^	1.2 ^a^	970 ^a^
		C_3_	3973 ^e^	0.99 ^d^	872 ^d^
		C_4_	349 ^g^	0.7 ^g^	98 ^g^
	30–60	C_3_ + C_4_	3408 ^o^		
		C_3_	3221 ^r^		
		C_4_	187 ^s^		
AS	0–30	C_3_ + C_4_	4101 ^a^	4.65 ^c^	1147 ^b^
		C_3_	3679 ^d^	3.75 ^f^	1005 ^f^
		C_4_	422 ^h^	0.9 ^h^	142 ^h^
	30–60	C_3_ + C_4_	3023 ^p^		
		C_3_	2785 ^q^		
		C_4_	238 ^s^		
UAN	0–30	C_3_ + C_4_	4190 ^a^	2.99 ^b^	1058 ^a,b^
		C_3_	3697 ^d^	2.46 ^e^	904 ^e,f^
		C_4_	493 ^h^	0.53 ^h^	154 ^h^
	30–60	C_3_ + C_4_	3269 ^o,p^		
		C_3_	3073 ^q^		
		C_4_	196 ^s^		

Within columns, values followed by the same letter (for topsoil: a–c for total amounts; d–f for C_3_ derived carbon; g–h for C_4_ derived carbon and for subsoil o–p total amount; q–r for C_3_, s–t for C_4_), are not significantly different (Tukey test, *p* < 0.05) between experiment plots. Number of replications *n* = 4.

**Table 4 plants-09-01217-t004:** Basic description of investigated location; C_SOM_—soil organic matter carbon.

GPS Coordinates	50°4′22″ N; 14°10′19″ E
Altitude (m above sea level)	410
Mean annual temperature (°C)	7.7
Mean annual precipitation (mm)	493
Soil type [32]	Haplic Luvisol
Soil texture [ 32]	Loam
Clay (%) (<0.002 mm)	5.4
Silt (%) (0.002–0.05 mm)	68.1
Sand (%) (0.05–2 mm)	26.5
Bulk density (g cm^−3^) topsoil [33]	1.47
Bulk density (g cm^−3^) subsoil [33]	1.55
C_SOM_ (%)	1.26
pH (CaCl_2_)	6.5
Cation exchange capacity (mmol_(+)_ kg^−1^)	118

**Table 5 plants-09-01217-t005:** Fertilizing design of the experiment; AS–ammonium sulfate, UAN–urea ammonium nitrate, UAN + st–urea ammonium nitrate + straw, FYM–farmyard manure, DM–dry matter, C–carbon, N–nitrogen.

Treatment	kg N ha^−1^ year^−1^	Organic Fertilizer (kg DM ha year^−1^)	C Content in DM (%)	C Supplied (kg ha^−1^year^−1^)	C Supplied during 26 Years (kg ha^−1^)	C/N in Org. Fert.
Control	—	—	—	—	—	—
AS	120	—	—	—	—	—
UAN	120	—	—	—	—	—
UAN + St	120 + 33.5 ^1^	5000	42.8	2140	55,640	79.3/1
FYM	120	5752	27.9	1603	41,678	13.4/1

^1^ N content in wheat straw; after application of UAN to UAN + St treatment the final C/N ratio changed to 14.6:1.

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
