# Peer review of "Soil Organic Matter Degradation in Long-Term Maize Cultivation and Insufficient Organic Fertilization"

_plants, 2020, doi:10.3390/plants9091217_

Round 1
Reviewer 1 Report
Dear authors
the manuscript presents a discrete amount of analytical data and the authors provide useful information on soil organic matter degradation in arable soil and report C losses in maize cultivation under different fertilization. Unfortunately the authors remain too attached to the analyzed territory giving their manuscript a regional and not very general interest.
The manuscript needs major revisions; the description on experimental design, the results and discussion should be reviewed. The conclusions of the manuscript should be more incisive and clear.
Unfortunately the paper is often unclear, the description of the results is confusing and they should be discussed better. The experimental plan is not well described and old and new data are confused. The results should be presented and discussed more carefully. In the description of the results there are some hints of discussion while In the first part of discussion, there is the simple results description. The discussion should have a general interest. Why is this work interesting? What is new?
The captions of the tables and figures must be more detailed; treatments, statistical differences, abbreviations, number of replicates and variability must be shown and described.
Specific point:
Abstract
Some sentences are unclear. For example Line 16: C4 ratio?
Keywords
The authors should modify some keywords, already present in the title.
Introduction
Lines 65: the Authors should add “Bettina et al.” before “[9]”;
Results and discussion
Perhaps dividing the results section into paragraphs might help reading. It would also be preferable to start with a description of the measured data and then move on to calculated data such as ratio. It is not always clear what data is actually collected in this experimental plan.
Line 93: where is this data shown?
Lines 141-146: Table 3???
Materials and methods
The “Sampling strategy” should be more clear. A crucial point is given by the number of sampling points and the replicates statistically analyze. The description of the experimental plan is confusing. Is the data collected in 1993? What data is measured after 26 years?
Lines 368-369: The sentence seems to report the results.
Figures and Tables
the captions should provide more detail. For example explain panels A and B. What do the numbers on the axes indicate? What do the different letters indicate? Which statistical test was carried out? What's the number of replicas? Are standard errors or standard deviation reported?
Author Response
Dear Reviewer 1,
Thanks for your valuable comments. Our detailed response to your remarks is in attached file.
Yours sincerely
Jiří Balík

Reviewer 2 Report
Very interesting article suitable for publishing.
Author Response
Dear Reviewer 2,
Thank you for the positive evaluating of our manuscript.
Yours sincerely
Jiří Balík
Reviewer 3 Report
Dear Authors,
Following are the observation of the manuscript.
Classification method for soil type is not presented.
Although, the results of soil texture and bulk density are presented in Table 4, the methods are missing.
Format of text in the tables should be uniformed.
Also, several notices and corrections are presented in the manuscript.
Thank you,
Best regards,

Author Response
Dear Reviewer 3,
Thanks for your valuable comments. We made all the changes according to your requests. We added the clasifications methods of soil type and soil bulk density in the Table 4. We uniformed text format (type and size) in the tables and we accepted
Yours sincerely
Jiří Balík
Round 2
Reviewer 1 Report
The paper is improved and the authors have followed many of the suggested indications. I have some doubts only about lines 17-19 of the abstract which are still unclear.
Author Response
Dear Reviewer 1,
Thanks for accepting most of the changes in our manuscript. According to your latest remark, we modified the sentence in abstract (rows 17-19) to be more clear.
Previous version was:
The weaker the soil extraction agent, the higher the ratio of carbon produced by maize was - this shows the lower stability of carbon compounds in maize stubble.
Actual version is:
With the weaker soil extraction agent (CaCl2) was determined 17.9-20.7% of carbon produced by maize. With stronger extraction agent (pyrophosphate) it was only 10.2-14.6%. It shows that maize produced mostly unstable carbon compounds.
Yours sincerely
Prof. Ing. Jiří Balík, CSc.
Round 3
Reviewer 1 Report
Accept in the present form